# Flexible PDMS-Based SERS Substrates Replicated from Beetle Wings for Water Pollutant Detection

**DOI:** 10.3390/polym15010191

**Published:** 2022-12-30

**Authors:** Chen-Hsin Lu, Ming-Ren Cheng, Sheng Chen, Wei-Lin Syu, Ming-Yen Chien, Kuan-Syun Wang, Jeng-Shiung Chen, Po-Han Lee, Ting-Yu Liu

**Affiliations:** 1The Affiliated Senior High School of National Taiwan Normal University, Taipei 10658, Taiwan; 2Department of Materials Engineering, Ming Chi University of Technology, New Taipei City 243303, Taiwan; 3Institute of Polymer Science and Engineering, National Taiwan University, Taipei 10617, Taiwan; 4Yottadeft Optoelectronics Technology Co., Ltd., Taipei 10460, Taiwan; 5Research Center for Intelligent Medical Devices, Center for Plasma and Thin Film Technologies, Ming Chi University of Technology, New Taipei City 243303, Taiwan; 6Department of Chemical Engineering and Materials Science, Yuan Ze University, Taoyuan City 32003, Taiwan

**Keywords:** surface-enhanced Raman scattering detection, water pollutants, bionic nanostructure, silver nanoislands, thermal evaporation

## Abstract

The flexible surface-enhanced Raman scattering (SERS) sensor, which has the bionic 3D nanoarray structure of a beetle-wing substrate (BWS), was successfully prepared by replicated technology and thermal evaporation. The bionic structure was replicated with polydimethylsiloxane (PDMS) and then silver (Ag) nanoisland thin films were deposited by thermal evaporation. The deposition times and thicknesses (25–40 nm) of the Ag thin films were manipulated to find the optimal SERS detection capability. The Ag nanoisland arrays on the surface of the bionic replicated PDMS were observed by scanning electron microscope (SEM), X-ray diffraction (XRD), and contact angle, which can generate strong and reproducible three-dimensional hotspots (3D hotspots) to enhance Raman signals. The water pollutant, rhodamine 6G (R6G), was used as a model molecule for SERS detection. The results show that 35 nm Ag deposited on a PDMS-BWS SERS substrate displays the strongest SERS intensity, which is 10 times higher than that of the pristine BWS with 35 nm Ag coating, due to the excellent 3D bionic structure. Our results demonstrate that bionic 3D SERS sensors have the potential to be applied in wearable devices and sensors to detect biomolecules and environmental pollutants, such as industrial wastewater, in the future.

## 1. Introduction

Surface-enhanced Raman spectroscopy (SERS) detection is a powerful analytical tool for trace-level detection with high sensitivity to characteristic vibrational fingerprints. SERS enhancement is divided into two main mechanisms for improving the SERS enhancement effect of the analyte. Firstly, the charge transfer between the analyte and the SERS-active metal substrate improves the signal; this is called chemical enhancement (CE) [1,2]. When the metal nanoparticles and the molecules are in direct contact, an adsorbate–surface complex is formed due to the electronic coupling effect of the molecules and the metal nanoparticles. Adsorbed molecules in this complex display larger cross-sectional effects than free molecules in typical Raman experiments. For example, some researchers synthesized low-dimensional semiconductor substrates, such as ZnO and GaN, to study the chemical enhancement. Moreover, using molecules such as 4-mercaptopyridine (4-MPY), 4-mercaptobenzoic acid (4-MBA), and 4-adenosine triphosphate (4-ATP) as analytes to measure SERS spectra can elucidate the charge transfer mechanisms between substrate and analyte molecules, leading to chemical enhancement. Hence, the charge transfers responsible for chemical enhancement have a mainly enhancing effect in SERS. Secondly, the metallic surface irradiated by a specific laser will generate an electromagnetic field, which then induces localized surface plasmon resonance (LSPR) [3,4] to enhance the signal; this is called the electromagnetic effect [5,6]. The electromagnetic effect originates from the plasmonic resonance of the conductive electrons on the surface of noble metal nanoparticles and the lasing light at the nanoscale, thus forming a huge local electric field on the surface, i.e., the surface plasmon. Surface plasmons are surface electromagnetic waves that exist at the interface of metals and dielectrics. Surface plasmon modes are localized around the surface of metal nanoparticles, creating a highly enhanced near field. According to the formation characteristics of surface plasmons, they can be divided into two categories: surface plasmon polariton (SPP) and localized surface plasmon (LSP). For example, the gold nanorods exhibit two well-divided LSPR bands. These bands have optical characteristics and high stability in solution, so gold nanorods can be conveniently used for the detection and characterization of absorbed molecules. Since SERS was discovered in the 1970s [7,8], it has had constant improvement along with the advancement of nanotechnology. However, traditional SERS sensor platforms were based on robust materials, such as glass [9], FTO glass [10], and silicon wafers [11]. There are also some different procedures of SERS substrate, such as nanoimprint lithography and sylgard 184. Ding et al. used composite nanoimprint lithography of etched polymer/silica opal films with electron-beam evaporation to fabricate a high-performance sensing substrate for UV-SERS [12]. The UV-SERS performance of DNA base adenine had been enhanced, revealing that it matched well with the UV plasmonic optical signatures and simulations. Moreover, Anindita Das et al. applied nanoimprint lithography on a flexible SERS substrate through electron-beam lithography (EBL), which was utilized as a master template, and the mold was later replicated via a nanoimprinting process to prepare a gold-coated polymer nanopyramid three-dimensional (3D) SERS substrate [13]. The intense electric field confinement at nanotips and four edges of a gold-coated polymer nanopyramid enhanced the Raman signal of probe molecules, which meant that rhodamine 6G, with a limit of detection down to 3.3 × 10^−9^ M, was achieved. Furthermore, Manuel Gómez et al. presented the potential of combining a plasmonic Bragg grating structure, which obtained a rough Al layer to tune SERS enhancement in the visible region [14]. The potential structure with excellent analytical reproducibility and amazing enhancement factors (10^7^–10^9^) are produced by the scalable procedure, which means it corresponds to the development of producing sustainable SERS substrates. Abeer Alyami et al. also fabricated some flexible, sensitive, and on-site detection-enabled substrates via Ag NPs/PDMS composites [15]. The Ag NPs/PDMS composites were obtained by self-assembly of Ag nanoparticle solutions on flexible PDMS surfaces. As a result, thiram concentrations and crystal violet (CV) within 10^−5^ M and 10^−7^ M were measured on fish skin and orange peel, which revealed the analytical versatility of SERS substrates. These SERS sensors are usually accompanied by complex fabrication techniques and numerous complicated procedures that are disadvantageous to the manufacturing stability of the product. These SERS sensors generally have high performance and sensitivity. Nevertheless, there are strong demands for the development of SERS sensors with high performance via facile and efficient fabrication processes.

Recently, flexible substrates have attracted enormous attention in the SERS field because of various advantages over rigid substrates, such as facile fabrication and shape control [12]. In contrast to conventional rigid substrates, flexible materials, such as polymers [16,17], papers [18], and nanofiber [19] films, have excellent flexibility that allows them to be attached to arbitrary surfaces for in situ detection. This property could further collect analytes, by pasting and peeling off from irregular surfaces, for rapid detection that avoids complex pretreatment. Moreover, bionic materials, such as butterfly wings, cicada wings, mantis wings, lotus leaves, and rose petals, have been widely considered to be excellent SERS substrates because of their 3D periodic microstructures. Moreover, there are studies with SERS substrates synthesized using a simple, low-cost, and environment-friendly method, wherein the component chitosan/chitin was utilized as an in situ reducer to synthesize gold nanoparticles in natural 3D photonic architectures [20,21,22]. The results show that a SERS substrate could detect 10^−9^ M of 4-ATP and exhibited the lowest relative standard deviation (RSD) with a moderate SNR [23]. Zhang et al. reported that the tip-based continuous, different dimensions of micro/nanostructure arrays were fabricated by the overlap of pyramidal cavities with different adjacent distances [24]. The 1362 cm^−1^ peak of Raman intensity of Au-coated PDMS substrate is about eight times higher than that of the commercial substrate. The SERS enhancement factor achieved an ideal level by using the nanostructured Au-coated PDMS surface. In the procedure of the PDMS substrate, the PDMS-BWS substrate displays a convenient way to replicate the nanostructure from the beetle wings, making the commercial biosensor possible in the future. Additionally, polymers such as PDMS have plasticity, and polyamide is most suitable for transfer to a bionic structure. Some studies have successfully transferred the structure of shark skin onto the outer layer of PDMS membranes to manufacture an ultrahydrophilic exterior for inhibiting bacterial adhesion. As a result, the use of PDMS not only inhibited bacteria and protected the wound, but also improved the hydrophilicity and biocompatibility of the wound repair [25].

In this study, polydimethylsiloxane (PDMS) was used to replicate the 3D structure of beetle wings to fabricate PDMS-BWS SERS substrates. The morphology, surface energy, and roughness could be tuned by depositing various thicknesses (25–45 nm) of silver (Ag) nanoisland thin films, and the SERS performance of the PDMS-BWS SERS substrates was evaluated by the water dye, rhodamine 6G (R6G). A novel, flexible, and reproducible bionic-nanostructure SERS substrate is demonstrated for the rapid determination of water pollutants, such as heavy metal, toxic molecules, and pesticides.

## 2. Experimental Section

### 2.1. Material

Beetle wings from *Anomalocera olivacea insularis* were purchased from a specimen store in Taipei. Ethanol and rhodamine 6G (R6G) were purchased from Aldrich Chemical Co (Milwaukee, WI, USA). Sliver slug was purchased from the Gredmann Group. SYLGARD™ 184 was purchased from Dow Corning.

### 2.2. Preparation of Beetle Wings

First, the beetle wings were immersed in acetone, ethanol, and deionized (DI) water, respectively, for 10 min under the ultrasonicator to remove surface impurities. Subsequently, the beetle wings were washed with DI water to remove the redundant solvents and dried in a circulator oven at 60 °C for 30 min.

### 2.3. Fabrication of PDMS-BWS-Ag SERS Substrate

PDMS-BWS-Ag was fabricated as a SERS substrate to rapidly detect water pollutant molecules through replication of the bionic 3D structure and deposition of Ag nanoislands. First, the silicone elastomer base reagent and the curing reagent were mixed to homogenization in a ratio of 10:1. The mixed reagent was poured on the beetle wings, removing bubbles in a vacuum, and then heated for 2 h at 60 °C until completely cured. After finishing the curing process, PDMS gel was split from the beetle wing to obtain the PDMS-BWS. As illustrated in Figure 1, PDMS-BWS-Ag was prepared from the PDMS-BWS by depositing Ag nanoislands with thermal evaporation at a rate of 0.5 Å/s and rotating at 10 rpm under high vacuum (10^−6^ Torr) to ensure homogenous deposition. Various Ag thicknesses (25, 30, 35, and 40 nm) were deposited onto the PDMS-BWS substrate, termed PDMS-BWS-Ag25, PDMS-BWS-Ag30, and PDMS-BWS-Ag35, respectively.

### 2.4. Characterization and SERS Detection

The 10 μL of R6G solution, with the concentrations of 10^−^^4^, 5 × 10^−^^5^, 10^−^^5^, 5 × 10^−^^6^, and 10^−^^6^ M, was dropped onto the PDMS-BWS-Ag substrate to measure the SERS spectra. The SERS measurements (632.8 nm He–Ne laser and 0.1 mW laser beam) were acquired by using a confocal Raman microscope (LabRAM HR Evolution, HORIBA France SAS, Kyoto, Japan) with a 50× objective lens with a detection range of 400–2000 cm^−^^1^. The contact-angle measurements were obtained using a contact-angle goniometer (DSA 100, Krüss GmbH, Hamburg, Germany). The XRD pattern was measured using a PANalytical-X’Pert PRO MPD operated at a scanning speed of 2°/min with a Cu Kα X-ray radiation (λ = 1.5405 Å). Scanning electron microscopy (SEM) (JEOL JSM-6701F) and an atomic force microscope (AFM) (Dimension Edge, Bruker, Berlin, Germany) were used to determine the morphology of the PDMS-BWS-Ag substrates.

## 3. Results

### 3.1. XRD Spectra

The crystalline structure of the PDMS-BWS-Ag SERS substrate was examined using XRD spectra. Figure 1 shows the XRD pattern of the PDMS-BWC-Ag SERS substrate, in which five diffraction peaks can be observed, corresponding to the 38.2° (111), 44.3° (200), 64.5° (220), 77.6° (311), and 83.3° (222) given by the Joint Committee on Powder Diffraction Standards (JCPDS, file numbers 04-0783 and 84-0713). These diffraction peaks were assigned to the crystalline nature of silver nanoparticles and demonstrate that the silver nanoislands had been successfully deposited onto the surface of the PDMS-BWC substrates.

### 3.2. Wettability of SERS Substrate

Figure 2 demonstrates the surface energies of the BWS-Ag, PDMS-BWS-Ag, and PDMS-Ag SERS substrates (Ag deposited thickness is 35 nm) using the contact-angle goniometer. The Ag deposited onto the pristine BWS (BWS-Ag) substrate was more hydrophilic, with a contact angle of 72.22° (Figure 2a). After transferring the BWS nanostructure to PDMS and then depositing the Ag (PDMS-BWS-Ag), the contact angle of the PDMS-BWS-Ag SERS substrate increased to 89.26° (Figure 2b), which was attributed to the more hydrophobic PDMS and the replica of the bionic nanostructure. Compared with the contact angle of the Ag deposited onto the pristine PDMS (PDMS-Ag, 81.72°, Figure 2c), the contact angle of the PDMS-BWS-Ag SERS substrate increased by 7.54°, which confirmed that the bionic (beetle wing) nanostructure has been replicated on the PDMS polymer gels.

### 3.3. Surface Topography Analysis

Figure 3 shows the SEM and AFM images of various thicknesses of a PDMS-BWS-Ag SESR substrate. Ag deposited onto the bionic PDMS-BWS by thermal evaporation forms a metal nanoisland film. The roughness of the PDMS-BWS-Ag SERS substrate increased with increasing thickness, which is similar to our previous findings [16]. According to the SEM image (Figure 3), Ag with a thickness of 25 nm deposited onto PDMS-BWS (abbreviated as PDMS-BWS-Ag25) demonstrated a relatively smooth surface, the root-mean-square roughness of which (Rq) was 0.44 nm (Figure 3d). In contrast, PDMS-BWS-Ag30 and PDMS-BWS-Ag35 exhibited a rougher surface by thickness increase, showing Rqs of 1.56 nm (Figure 3e) and 1.86 nm (Figure 3f), respectively. The results show that the Ag nanoislands successfully immobilized on the bionic beetle-wing nanostructure. The rougher nanostructure was beneficial to generating higher localized surface plasmon resonance (LSPR) effects, which obtain a higher Raman enhancement factor. The Raman enhancement analysis is discussed in detail in Figure 4.

### 3.4. SERS Spectra

The various thicknesses (25–40 nm) of Ag deposition by thermal evaporation influence the extent of the SERS enhancement. Here, the water dye of R6G was used as a model molecule. According to a previous report [26,27,28], the characteristic SERS peaks of R6G are 612, 1361, 1510, and 1649 cm^−^^1^, which are assigned to C-C-C ring in-plane bending, C-H in-plane bending, and aromatic C-C stretching, respectively [29]. Figure 4 shows the SERS spectra of R6G with varied thicknesses of the PDMS-BWS-Ag SERS substrate. The R6G characteristic peak located at 612 cm^−^^1^ was used as the characteristic indicator to evaluate the SERS enhancement. The results show that the SERS intensity increased in relation to the increasing thickness of Ag deposition from 25 to 35 nm (Figure 5). However, the SERS intensity was significantly reduced when the thickness was 40 nm. This suggests that the saturated Ag nanoparticles were coated on the PDMS-BWS substrate, causing a reduction in SERS intensity. Comparing the results, it could be inferred that 35 nm of Ag coating (PDMS-BWS-Ag35) the SERS substrate demonstrates the highest SERS signals. Therefore, the following experiment utilized PDMS-BWS-Ag35 as a SERS substrate to detect the different concentrations of R6G. The relation between the Ag thin film thickness and the Raman intensity of R6G can also be verified in the reference [30].

Herein, the change in SERS intensity was investigated with various SERS substrates after the deposition of 35 nm Ag nanoislands, i.e., PDMS-BWS-Ag35, BWS-Ag35, PDMS-Ag35, and glass-Ag35 (Figure 6). A 10^−5^ M R6G solution was dropped onto the different SERS substrates to evaluate their SERS enhancement. The results indicate that the order of SERS intensity is PDMS-BWS-Ag35 > BWS-Ag35 > PDMS-Ag35 > pristine glass-Ag35 SERS substrates. The pristine glass slide with 35 nm Ag nanoislands (glass-Ag35 SERS substrate) showed a very weak (almost absent) SERS signal. However, a stronger SERS signal with R6G was displayed using PDMS-Ag-35 due to the rougher surface structure compared to the glass slide. Furthermore, it was also possible to detect an obvious SERS signal with a 35 nm Ag coating on natural beetle-wing substrates (BWS-Ag-35 SERS substrate). Nevertheless, the SERS intensity increased ~10 times with the PDMS-replicated bionic beetle-wing substrate coated with 35 nm Ag nanoislands (PDMS-BWS-Ag35 SERS substrate). This means that PDMS with a beetle-wing nanostructure could create the optimal hotspot effect to induce the strongest SERS intensity.

For limit of detection (LOD) measurement, various concentrations of the R6G solution were measured using the PDMS-BWS-Ag35 SERS substrate. The SERS intensity of the characteristic peaks (612 cm^−1^) progressively decreased until it reached a concentration of 10^−6^ M (Figure 7). It should be noted that the SERS intensity decreased when the concentration approached 10^−4^ M, which can be attributed to the decreasing induction of the SERS signals due to R6G molecular aggregation. We tried to calculate a calibration curve for the PDMS-BWS-Ag35 SERS substrate using various concentrations (5 × 10^−5^~10^−6^ M) of R6G molecules as a function of the SERS intensity (integrating the area of the characteristic peaks, 612 cm^−1^) in Figure 8. The result shows a good linear relationship, suggesting that the calibration curve exhibits high reliability. Although the result of the LOD measurement compared with other similar research [31] was not good enough, this unique 3D bionic structure demonstrates the potential possibility of SERS detection. In addition, the PDMS-BWS substrate has more flexibility within the advantages of being an environment-friendly and easy procedure than the cicada wings.

In addition to using LOD analysis to evaluate the SERS performance of the PDMS-BWS-Ag35 SERS substrate, we utilized the following (Equation (1)) to calculate the enhancement factor (EF) of Raman spectroscopy:(1)$$\mathrm{EF}=\left(I_{\mathrm{SERS}}/N_{\mathrm{SERS}}\right)/\left(I_{\mathrm{Raman}}/N_{\mathrm{Raman}}\right)$$
where I_SERS_ and I_Raman_ are the intensity of the SERS spectra and Raman spectra, respectively, for R6G molecules at 612 cm^−1^. N_SERS_ and N_Raman_ are the numbers of R6G adsorbed on the SERS and blank substrates, respectively. According to Figure 6 and Figure 7, the EF of the PDMS-BWS-Ag-35 SERS substrate was 1.9 × 10^4^ in optimized conditions.

Therefore, the proposed technique of PDMS replication and Ag deposition by thermal evaporation allows us to achieve SERS intensification through the utilization of LSPR plasmonic coupling phenomena and the generation of a huge electric field; this enhancement is attributed to the nanostructure of the bionic beetle wing replica.

## 4. Conclusions

We successfully developed a novel bionic nanostructure replication technique from beetle wings, with deposited Ag nanoislands, as a SERS substrate (PDMS-BWS-Ag) to detect water pollutants. Comparing with other references of nanoisland substrates (the dragonfly wings [32]), the beetle wings display unique and natural nanostructures to produce enormous hotspots for Raman enhancing. In addition, the BWS-PDMS-based substrate exhibits a flexible, environment-friendly, and easier to produce procedure, which makes the chance for the commercialization of the products. Subsequently, the SERS performance is optimized through the deposition of different thicknesses of Ag nanoislands by facile thermal evaporation methods. The results indicate that PDMS-BWS-Ag35 demonstrates excellent SERS enhancement performance, which is 10 times stronger than pristine BWS-Ag35 (without the replication process). The LOD of PDMS-BWS-Ag35 SERS is lower than 10^−6^ M and displays a linear calibration curve from 5 × 10^−5^~10^−6^ M. These results illustrate that the PDMS-BWS SERS substrate has the potential to be applied in detecting biomolecules and water pollutants, such as adenine from DNA, bacteria, viruses, water toxicity, and water heavy-metal ions.

## Data Availability

The data presented in this study are available on request from the corresponding author.

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
