# Peer review of "Flexible PDMS-Based SERS Substrates Replicated from Beetle Wings for Water Pollutant Detection"

_polymers, 2022, doi:10.3390/polym15010191_

Round 1

Reviewer 1 Report

The manuscript entitled "Flexible PDMS-based SERS Substrates Replicated from Beetle Wings for Water Pollutants Detection" by T.-Y. Liu et al. describes the application of SERS sensor on an innovative PDMS/ Ag thin films substrate in detection of organic dyes (R6G).

The idea of utilization of a 3D bionic flexible structure as substrate for SERS sensors is innovative, it should be better highlighted in Introduction.  The experimental results are important in the field of SERS sensors. There are some issues to be resolved:

1. in Abstract the authors claim that silver thin films are deposited on the surface of PDMS-BWS, but at line 19 they also mention the use of Au. Please clarify the information in this section.

2. Introduction section is in the state-of-the-art, there are some abbreviations in lines 38-39 to be explained.

At lines 63-81 some literature data regarding SERS based on PDMS need to be added, especially those referring to the detection of R6G, highlighting more the novelty of this paper.

3. In section 2.1., as well in Abstract, the authors mentioned the fabrication of Ag nano islands. Please add an AFM image to prove their formation.

The Section 2.1 is twice. Please specify at line 113 the concentration of R6G in 10 uL.

4. In Section 3, in Fig. 1, the comparative XRD patterns of the substrate before and after Ag deposition must be added.

At line 170, change "the water dye of the rhodamine"

In Fig. 4 the concentration of R6G is 10-4M. The reference to this figure should be added in the comments at lines 210-211. Possibly to add both graphs (Fig. 4 and Fig. 6) in one figure to better highlight.

The detected concentrations of R6G is higher than in other studies. Please add some literature comparisons in this section.

Based on these comments my recommendation is Major Revision.

Reviewer 2 Report

The manuscript by Liu and colleagues concerns the fabrication of novel and flexible SERS substrates for the detection of water pollutants.

The topic is interesting, but in order to make it appealing for this journal (and why they selected this journal) they should emphasize the importance/advantages of using a polymer for the fabrication of such substrates. More in general, the following are the changes I suggest to improve the overall quality of the manuscript.

- Introduction: even though many readers interested in SERS possibly do not need any explanation, the introduction must introduce SERS and the concept of SERS sensors, especially for the average readers. Please try to explain in a more fluent and precise way the concept of SERS, the different mechanisms that generate such enhancement, and also the meaninghcaracteristics of SERS sensors. In addition, please try to cite the articles already published on strictly related examples of SERS uses or substrates fabrication. As an example, it is possible to find works on the use of nanoimprint lithography to produce SERS substrates (since Nanoimprint Lithography of Al Nanovoids for Deep-UV SERS Tao Ding, Daniel O. Sigle, Lars O. Herrmann, Daniel Wolverson, and Jeremy J. Baumberg, ACS Applied Materials & Interfaces 2014 6 (20), 17358-17363), some of them even using sylgard 184 (M Gomez, S Kadkhodazadeh, M Lazzari, Surface enhanced Raman scattering (SERS) in the visible range on scalable aluminum-coated platforms, Chem. Commun., 54 (2018), pp. 10638-10641) to produce flexible SERS substrates (A. Alyami, A.J. Quinn, D. Iacopino, Flexible and transparent surface enhanced Raman scattering (SERS)-active Ag NPs/PDMS composites for in-situ detection of food contaminants, Talanta, 201 (2019), pp. 58-64; Das, A., Pant, U., Cao, C. et al. Fabrication of plasmonic nanopyramidal array as flexible SERS substrate for biosensing application. Nano Res. (2022). https://doi.org/10.1007/s12274-022-4745-0).

In addition, the end of the introduction should introduce the element of novelty of the author's approach. At the same time, in the conclusions section, the authors should highlight the main advantages/differences between the results they obtained and those from similar, already published, works.

 -in the section 3.4, the authors have to compare their results, with those reported in other articles (as an example, those they decide to cite in the introduction section, or others), to highlight whether the enhancement factor they obtained are competitive or which could be the reasons of such differences.

Round 2

Reviewer 1 Report

The authors fully responded to all observations. The manuscript has been improved considerably. I recommend the publication of this paper in the present form.

Reviewer 2 Report

it may be accepted in the present form